# Newborn Screening Is on a Collision Course with Public Health Ethics

**DOI:** 10.3390/ijns8040051

**Published:** 2022-09-26

**Authors:** Robert J. Currier

**Affiliations:** Department of Pediatrics, University of California, San Francisco, CA 94158, USA; robert.currier@ucsf.edu; Tel.: +1-778-887-9581

**Keywords:** newborn screening, ethics, consent, DNA sequencing, genome, genomic newborn screening, Krabbe disease, Pompe disease, adrenoleukodystrophy

## Abstract

Newborn screening was established over 50 years ago to identify cases of disorders that were serious, urgent, and treatable, mirroring the criteria of Wilson and Jungner. In the last decade, conditions have been added to newborn screening that do not strictly meet these criteria, and genomic newborn screening is beginning to be discussed. Some of these new and proposed additions to newborn screening entail serious public health ethical issues that need to be explored.

## 1. Introduction

Newborn screening was named by the Centers for Disease Control among the Ten Great Public Health Achievements—United States, 2001–2010 [1]. This decade brought a four-fold increase in the number of screened disorders and the development and adoption of the Recommended Uniform Screening Panel (RUSP) by the Advisory Committee on Heritable Diseases of Newborns and Children (ACHDNC) [2].

Currently newborn screening programs are set up to collect a specimen and send it to the newborn screening laboratory within 48 h of birth, with results returned to the pediatrician, and often a specialist, shortly thereafter. There is no parental consent, although some States allow an “opt out” for religious or personal reasons. On the contrary, the perinatal period is a time when most parents are recovering from birth, consumed by the overwhelming adjustments of parenthood, and excited about the new baby. It is not an opportune time for a detailed discussion of rare events and formal consent. 

However, public health newborn screening without formal consent requires justification based on the urgency, severity, and treatability of the diseases targeted [3]. In order for the State to supersede the decision-making authority of parents over the health of their child, the essential argument has been that irreparable harm will be done if the screening does not proceed immediately. This points to both the seriousness and urgency of the screened conditions. Thus, after identification of an affected infant, it is necessary that effective treatment is available that can avert otherwise dire consequences. For this reason, identification of untreatable conditions has never been accepted by newborn screening programs as a goal of newborn screening. The constellation of these factors—seriousness, urgency, and treatability—provides a compelling justification; the vast majority of parents would consent, were they to fully consider the process [4].

Previous studies have considered newborn screening from the perspective of medical ethics or medical economics. Here, we will discuss newborn screening from the perspective of public health ethics within governmental ethics.

## 2. Discussion

### 2.1. Public Health Ethics as a Framework for Consideration of Newborn Screening

The State has many roles in conducting a newborn screening program. In addition to providing laboratory testing and interpretation of results or contracting with professionals to do so, the State provides education to medical providers and to parents, coordinates follow-up and provides payment for services in many newborn screening programs, conducts epidemiological monitoring to better understand disease incidence and natural history, regulates the entire newborn screening system, and makes sure that newborn screening is provided for all newborns.

As such, the ethical evaluation of newborn screening can appropriately be located within governmental ethics, and particular, public health ethics. The principles of public health ethics focus primarily on improving population health overall, including health equity, reducing disparities, and removing societal barriers to health. In all of these activities, individual autonomy may be limited, which requires a separate ethical framework for analysis. Cass [5] has organized this framework as a series of six sequential questions that any proposed public health project should be able to answer with supportive data. Responses to these questions with regard to newborn screening for phenylketonuria (PKU, MIM: 261600), the paradigm of newborn screening tests, show how these principles apply. 

**What are the public health goals of the proposed program?** The goal of newborn screening for PKU is the early detection of PKU to enable treatment (dietary modification) to avoid irremediable damage to the newborn’s brain.**How effective is the program in achieving its stated goals?** Newborn screening is applied universally, so its reach is the entire population of newborns. Screening for PKU has been highly effective at identifying cases, to the point that any reported case missed by screening requires an investigation of causes and a plan for remediation.**What are the known or potential burdens of the program?** A universal burden of newborn screening is the loss of parental autonomy with regard to the medical care of their child. Secondary burdens are those relating to positive results caused by other factors than PKU itself. These off-target results require follow-up and a period of uncertainty and anxiety in parents until PKU is ruled out.**Can burdens be minimized? Are there alternative approaches?** Newborn screening without parental involvement is considered unavoidable in order to maintain the universality of the program. Two developments over the course of the decades of screening for PKU have greatly reduced the burdens of off-target results. First, the existence of hyperphenylalaninemia has been recognized, and infants identified with this condition are not required to restrict their diets as severely. Second, the screening test itself has become much more specific, first by the addition of the ratio of phenylalanine to tyrosine, and later by interpretation of a much larger panel of newborn screening markers. These developments have greatly reduced the number of off-target results.**Is the program implemented fairly?** As indicated previously, newborn screening has always been implemented as a universal program. All newborns are tested, without regard to birth location, insurance coverage status, or ability to pay.**How can the benefits and burdens of a program be fairly balanced?** In general, newborn screening for PKU has achieved a very high degree of recognition and satisfaction for the real benefits that it has achieved. The loss of parental autonomy is considered by some to be a reason for eliminating newborn screening entirely, but that is very much a minority opinion.

Some recent additions to newborn screening highlight how problematic some of these principles have become. 

### 2.2. Three Problematic Recent Introductions to Newborn Screening

As newborn screening has evolved and has moved farther from the original principles of Wilson and Jungner [3], disorders have been added to programs that raise issues from the perspective of public health ethics. We look at three specific screened disorders, then apply these public health ethics questions to them. Afterward, we consider the response of some newborn screening programs to these issues.

#### 2.2.1. Krabbe Disease Screening in New York

The Newborn Screening Program of New York State was mandated by the New York legislature to begin screening for Krabbe disease, a lysosomal storage disorder (KD. OMIM:245200) in August, 2006. Orsini et al. reported on the first eight years of screening in 2016 [6]. They found, that of 2,090,910 infants tested, there were 10,199 that required retesting to confirm low enzymatic activity. From these, 620 were reflexed to molecular testing. Within that group, 272 had only benign polymorphisms in the *GALC* gene and 348 were referred for follow-up. 

The clinical outcome of these cases identified by screening was reported by Wasserstein et al. [7]. Within the group of 348 infants referred for follow-up, the diagnostic testing found that 203 were at no risk for infantile KD, 92 were at low risk, 37 were at moderate risk, and 14 were at high risk, and 5 of those at high risk had confirmed early infantile Krabbe disease (EIKD). The confirmatory testing included extensive neurological evaluation and measurement of protein in CSF. These 5 families were offered hematopoietic stem cell transplantation (HSCT). One family refused; based on the genetic result for the infant they anticipated that treatment would not be successful. The other four infants were transplanted, between 24 and 41 days of life. Of these infants, two died of transplantation-related complications. The infants who were identified as high-risk but not EIKD were followed for 1 to 9 years, with no symptoms of Krabbe disease.

Ehmann and Lantos [8] considered these data pointed to the positive predictive value of the screening test of only 1.4%, which they characterized as too inefficient. A more recent publication from New York [9] explains the current screening algorithm which involves evaluating a panel of enzyme activities and other newborn screening markers. From a population of roughly 260,000, the number of referrals was reduced from 48 to 10.

An essential tool of this more precise algorithm is the use of the Collaborative Laboratory Integrated Reports (CLIR, Mayo Clinic, Rochester, MN, USA) tools, which use data submitted from multiple programs around the world for affected and unaffected newborns. Combining multiple newborn screening analytes, in this case a panel of enzymes implicated in lysosomal storage diseases, can provide this better separation of high-risk infants warranting referral for diagnosis from other types of results.

Within the group of high-risk infants referred for diagnosis in the New York study, almost two-thirds were ruled out from EIKD. This points to an additional need to refine the screening test to be able to distinguish these subgroups of high-risk infants. Recent work suggests that psychosine is very highly elevated in the EIKD group, and much lower in the rest of the high-risk group. Implementing psychosine testing as a second-tier test within the Krabbe screening algorithm can greatly improve the overall screening performance [10].

Considering the public health ethics framework questions, the question how effective the program is at meeting its goals already points to potential problems. The goal of the screening test is to diagnose and effectively treat EIKD. In the original implementation, there were very many more off-target results than true positive results. Although a more recent publication from New York shows that an alternative algorithm for the screening test reduced the burden of these off-target results substantially, the data on the effectiveness of treatment has been called into question as pointed out by Ehmann and Lantos. Observing these results from New York, several newborn screening programs have declined to implement screening for Krabbe disease. In 2009, the ACHDNC considered screening for Krabbe disease and did not recommend screening based on gaps in evidence. In the intervening years, additional data and improvements in the screening algorithm suggest that reconsideration may be warrented.

#### 2.2.2. X-Linked Adrenoleukodystrophy

Newborn screening for X-linked adrenoleukodystrophy [X-ALD] was added to the RUSP in 2016. X-ALD (MIM:300100) is an X-linked recessive disorder. Like Krabbe disease, X-ALD has a variable phenotypic spectrum, from childhood onset cerebral adrenoleukodystrophy, which is the target of newborn screening, to later onset forms, including isolated adrenal insufficiency without central nervous system involvement. 

The evidence review prepared for ACHDNC was definitive:
It is estimated that about 20% of heterozygote females have VLCFA plasma levels within normal limits. However, because females with X-ALD do not typically experience symptoms until adulthood, if ever, they are not a target of newborn screening.[11] (p. 5)

Nonetheless, screening for X-ALD includes the entire population, males and females. Advocates of screening point to benefits of screening females, including the potential to identify females who might experience symptoms later in life, and the indirect identification of mildly affected older siblings or other male relatives who had been previously undiagnosed. 

In newborn screening for X-ALD, the benefits accrue almost entirely to males, and the burdens primarily to females. This suggests that the program as implemented is unfair, and alternative approaches should be considered. In this case, the need for immediate treatment is lower, so it is possible to consider a screening test offered somewhat later by the pediatrician. This would also allow time for discussion and reflection on the part of the parents.

The newborn screening program in the Netherlands decided to go forward with a boys-only newborn screen for X-ALD with a four-tier algorithm. To avoid errors in recording the sex on the heel-prick card, the screening algorithm includes a count of X-chromosomes. Only infants with a single X-chromosome go on to further tiers of screening [12].

#### 2.2.3. Pompe Disease

Pompe disease (MIM:232300) is another lysosomal storage disorder; it was added to the RUSP in 2013. Like the above conditions, Pompe disease has both infantile-onset and late-onset forms, with approximately 28% being infantile-onset. The classical infantile onset form includes cardiomyopathy, a primary cause of neonatal morbidity in this condition.

The screening test as initially implemented had a high false positive rate and consequently a low positive predictive value. More recent research has shown that a larger panel of newborn screening analytes combined with the CLIR tools for lysosomal storage disorders is better able to distinguish between the true positive cases from the false positive/pseudodeficiency cases [13].

Screening for Pompe disease has also highlighted the reality for individuals with late-onset disease identified as newborns [14]. There is no consensus on the clinical care for these patients—what tests to order and how to interpret them, what treatments to consider and when to begin them. Essentially, newborn screening enrolls these patients in an unconsented clinical trial of case management. This cannot be considered a valid goal of newborn screening.

### 2.3. Genomic Newborn Screening Requires Multiple Ethical Considerations

Over 400 genetic conditions have been identified [15] as meeting the initial screening criteria: pediatric onset, some level of severity, and some ability to intervene. Within this group, there is a smaller subset of disorders with newborn onset that are serious and highly actionable. Some of these, in the absence of any other testing method, could be the targets of a DNA sequencing-based newborn screening test. The goal of this test, like all newborn screening, would be the identification of affected infants and the early treatment of the condition to prevent adverse consequences. It is unlikely that such a screening test could be implemented effectively. The DNA sequence results identify variants in the gene, but the inference of the possible disease state—whether early-onset, late-onset, or not penetrant—is difficult to predict for some conditions. One possibility is to restrict the group of target conditions still further to conditions that have well-developed genotype/phenotype correlations.

In addition, it will be difficult, if not impossible, to implement genomic newborn screening in a way that treats the population equitably. The interpretation of the clinical significance of the variants detected by sequencing relies on genomic databases. However, the existing databases of genetic variants overrepresent the variants found in individuals of European ancestry and underrepresent the variants in individuals of other ancestries. The consequence of this divergence is an increased likelihood of finding a variant of uncertain significance (VUS) in individuals of underrepresented ancestry. Whether VUSs are released to the patient or not, the implication is that the performance of genomic newborn screening will be different for different populations. This issue in particularly important in parts of the United States with large immigrant populations from Mexico, Central and South America. Due to the expense of genetic testing in their home countries, these individuals tend to be underrepresented in genetic databases.

One burden of genomic newborn screening is structural within the public health department. Genomic sequencing is currently more than two orders of magnitude more expensive than any current newborn screening test. Budgets for newborn screening are not limitless, and fees cannot be raised arbitrarily without limit. The introduction of genomic newborn screening has the potential to divert resources from other responsibilities, including follow-up, diagnostic testing, and treatment. The expense of genomic sequencing also has the potential to absorb funds that the larger public health department might need for other priorities. The scale of the funding involved could imply a significantly more convoluted contract management process, with additional levels of State supervision beyond the newborn screening program. 

Another population-wide burden arises from sequence data as intensely personal data for the newborn, but which also has implications for the parents. This challenge to trust will require the newborn screening program to practice extreme transparency in how the DNA is used, whether residual DNA is stored, how the sequence data is generated, and how the results will be safeguarded for the future. The possibility of future use of the data or the DNA for the benefit of the individual (by reinterpretation of variants) or for more general public health benefit (from research) must be explicitly communicated. Potential harm to the same person might arise from law enforcement requiring access to DNA or data for future forensic identification purposes of the person or relatives. 

The clear alternative is to undertake genomic newborn screening only after detailed discussions with parents about the kinds of results and the implications for diagnosis and treatment. Parental preferences will need to be elicited, recorded, and complied with.

Failure in any of these areas of public health ethics has the potential to dramatically undercut the public’s trust that the newborn screening program is acting in the public interest. This increased distrust is one precondition for the crisis to come.

### 2.4. The Coming Storm

A significant threat to the future of newborn screening is political polarization driven by social media. It is plausible that the social media environment is primed to attack genomic sequencing in newborn screening. Any online discussion of universal DNA sequencing will attract the single-word comment “GATTACA”, referring to the dystopian film from 1997 about a eugenic future of genetic determinism. It is easy to imagine that the announcement by a single State of genomic newborn screening without parental involvement will catalyze a group of privacy absolutists who begin with a deep suspicion of governmental activity to mount an online campaign against all newborn screening. With the example of the social media campaign against vaccination for SARS-CoV-2, we can see how a small number of highly influential individuals created dissension that was magnified by millions of followers and amplified by social media algorithms that prioritize controversy and outrage. In addition, there are groups of malign actors who seek not to promote policy, but to foment distrust of government and discord in the population. 

The threat of political polarization can already be seen in the controversy over the storage and future use of residual dried blood specimens from newborn screening that are obtained without explicit consent for storage or use, exemplified by the lawsuit Kanuszewski vs. Michigan Department of Health and Human Services, discussed below.

The controversy over dried blood spots points to a second, and potentially more serious, threat to the future of newborn screening: a constitutional challenge to the State’s ability to do newborn screening at all. A recent case from Michigan, Kanuszewski vs. Michigan Department of Health and Human Services, was heard by the United States Court of Appeals, Sixth Circuit. This complicated case involves allegations that rights of (a) the newborn and (b) the parents were violated with regard to: (1) the collection of the specimen and (2) the storage and future analysis of the residual dried blood spot. In the Court’s ruling, the analysis of standing and the analysis of the claims regarding the newborn and the parents were separated into four combinations. When the Court considered the claim that the newborn’s rights had been violated by the specimen collection [ax1], the Court first noted that the State’s sovereign immunity precluded the claims. However, the Court further noted that the newborn does not have a right to direct their own medical care, so that there was no constitutional violation. However, when the Court ruled on the parents’ claim [bx1], the State’s sovereign immunity again precluded the claim, but the Court declined to go further into an analysis of the constitutional issue:
In contrast with the issue discussed in the previous section [RJC: regarding the rights of the newborn], we cannot easily say, based on the allegations in the Complaint, that the drawing of the children’s blood “do[es] not make out a constitutional violation” of the parents’ substantive due process right to direct their children’s medical care. *Id.* The Supreme Court has suggested that we might decline to exercise our jurisdiction in situations where “it is plain that a constitutional right is not clearly established but far from obvious whether in fact there is such a right”. *Pearson*, 555 U.S. at 237, 129 S.Ct. 808. Because this issue presents such a situation, we decline to rule on whether the initial drawing of blood violated the parents’ substantive due process rights.[16] (p. 416)

This ruling points to the thinness of the thread that supports newborn screening. Another court, in another jurisdiction, may choose to go further and rule that the collection of the newborn screening specimen is an unconstitutional violation of the Fourth and Fourteenth Amendments. A ruling to that effect would require immediate and radical restructuring of newborn screening, with the possibility of a significant hiatus during the restructuring process resulting in significant irreversible but preventable harm to infants in the short term, and, if the mandate of newborn screening for all were eliminated, in the long term as well. 

### 2.5. Sustaining Newborn Screening Will Require Challenging Adjustments

#### 2.5.1. Establish, Maintain, and Monitor a Very High Standard for the Positive Predictive Values of Newborn Screening Tests

The genetics workforce is under increasing pressure to do more. Furthermore, the number of new genetic counselors, clinical geneticists and medical geneticists being produced is not sufficient to meet the demand. False positive results from newborn screening can produce a larger burden in diagnostic work-up than true positives, because it is a matter of ruling out all possibilities, a process that can require multiple visits to specialists over months to years. As stewards of the human resources of the follow-up programs, newborn screening has the responsibility to avoid excessive false positive results. Each screening test should aim for fewer false positive results than true positive, i.e., for a PPV greater than 50%. The examples of the use of CLIR tools for both Krabbe disease and Pompe disease show that this benchmark can be achieved. It does require a collaborative effort to collect and unify the world-wide data on these rare diseases. Additionally, second-tier genetic testing (either mutation panels or sequencing) applied to a selected group of genes is currently being used in NBS programs to reduce false positive rates. Some of the issues of genetic testing cited above do not apply, principally because the first-tier positive NBS test provides an initial phenotype and indication that further testing is warranted.

#### 2.5.2. Implement Only Screening Tests That Can Single Out Early Onset Forms of a Disorder to Avoid Reporting Late-Onset and Non-Penetrant Forms

Of all results other than true target disorders, late-onset forms create a substantial, and ongoing burden for the follow-up center. The infant will need to be followed periodically to determine if or when clinical indications of disease may appear, and then to plan and monitor an ongoing treatment regimen.

In addition, the parents of infants with a late-onset disorder identified by newborn screening could justifiably feel that their consent should have been obtained before the screening went forward. The larger this cohort becomes, the greater possibility that a group of parents will decide to sue the State to end the practice.

#### 2.5.3. Include in Newborn Screening Only Disorders That Have Serious, Irremediable Consequences within the First Weeks of Life

The only defense that the State can offer to a parental suit alleging a due process violation in the failure to obtain consent for newborn screening is for the State to assert that screening was a matter of preventing death or severe disability for the infant. If the designated follow-up for a disorder identified by newborn screening is watchful waiting and periodic monitoring for the first year of life or more, this defense would not apply, and the suit could prevail.

In the event that the Court rules that screening for a particular disorder without consent is not permitted, the transition to removing that screening test could cause disruptions to the overall screening process. In some LIMS systems, the removal of a screening test would need support from the LIMS vendor. At the same time, parents of infants who were not identified as having the condition could feel that the State did them harm, which might also result in legal liability for the State.

#### 2.5.4. Develop a Life-Course Staged Approach to Public Health Genetic Screening

Disorders that do not meet the urgency criterion above may still be important targets for public health screening. Other medical screening tests, for example colonoscopy, have a schedule of examination tailored to the individual’s medical history, including family history, and previous testing. Genetic screening needs to expand the time frame to allow for more complex discussions of what testing will entail, what kinds of results may arise, and what follow-up may ensue. As the child becomes older, they may also have a perspective that needs to be included.

The challenge will be to institute expanded screening beyond the newborn period in a way that makes it available to all, in the same way that newborn screening is. Advocates hoping to include additional disorders in newborn screening underline the importance of identifying all affected infants. A similar assurance that screening will be available to all will need to accompany genetic screening tests carried out later in life, while recognizing that some families may choose not to participate. The Early Check program in North Carolina has begun to explore later newborn screening with consent. This program grew from a desire to implement early screening for Fragile-X syndrome, while recognizing that the testing is unsuitable for the standard newborn screening system [17]. As an additional example, the State of California has made prenatal screening for certain birth defects available to all residents of the State who consent to have it, originally through serum biomarker testing of a maternal blood sample and ultrasound, more recently by analysis of cell-free fetal DNA in maternal blood (noninvasive prenatal testing or screening, referred to as NIPT or NIPS).

#### 2.5.5. Alternatively, Find a Way to Inform Parents about Newborn Screening and Get Genuine Consent

Consent for newborn screening has long been resisted, in the belief that the consent process is too burdensome and poorly comprehended during the period immediately following the birth and that the impact of parental refusal to consent will be primarily borne by disadvantaged populations. There has been substantial research on the effectiveness of prenatal education about newborn screening and about the use of the residual DBS [18]. If the transition from prenatal care to neonatal care were smoother, it might be possible to do much of the education and consent for newborn screening during prenatal care.

## 3. Conclusions

The issue of consent for newborn screening seems to be fundamental for its future. It seems likely that genomic sequence data will be determined to be Protected Health Information (PHI) as being inherently identifiable. Such a determination is likely to increase pressure on NBS programs that use sequence data to get consent for its generation and storage. New screening tests are likely to take a longer time for development and refinement to meet screening standards. This will imply an extended period of pilot testing, at first with consent; then later, perhaps, without.

Newborn screening has been a tremendously successful public health program. However, the future of newborn screening is not guaranteed. We should not go forward without considering whether what we do today will make its survival more or less likely.

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
