# Peer review of "Newborn Screening Is on a Collision Course with Public Health Ethics"

_2409-515X, 2022, doi:10.3390/ijns8040051_

Round 1

Reviewer 1 Report

Suggestions for the author:

Line 23 …pediatrician, and often a specialist, shortly thereafter

Line 45.  interpretation of results, or contracting with professionals to do so,

Line 46-47. provides payment for services in many states

Line 115.  According to the reference, 14 were determined to be at high risk, 5 of whom were confirmed to have EIKD. 4 went to transplant, one family declined and the remaining 9 were followed from 1 to 9 years and hadn’t yet developed symptoms. 

Line 123. The reasoning here is not clear. Why does the fact that only 5 of 14 high risk infants were determined to have EIKD suggest that some indeterminate part of the group that got HCST pre-symptomatically might never have gone on to develop EIKD, even without intervention? This might better be suggested by the data cited above regarding symptom-free follow-up for 1-9 years of the remained of this high risk cohort. However, there is no  inclusion of the explanation of why only 5 of the 14 were actually diagnosed with Krabbe. From Wasserstein, those 14 infants underwent extensive neurodiagnostic evaluation as appropriate for follow-up of a positive newborn screening test. In this case, only 5 were confirmed to have Krabbe by standard-of-care diagnostic testing, and thus were candidates for HSCT. This might obviate the argument that is being made about effectiveness of treatment. Refinements in screening, including addition of psychosine, also warrant specific mention.

Line 177: Saying that over 400 disorders meet basic newborn screening criteria ignores the nuances of the  Milko, et al paper and overstates their conclusion. The age range for these conditions is birth to age 18 (pediatric) with a broad range of interventions including surveillance. Not all have the immediacy of intervention needs like most of the RUSP conditions.

Line 185 a reflection on the possibility of adding sequencing only for those conditions that have clear genotype:phenotype correlation would be appreciated rather than a blanket ruling out of this approach. The commentary seems to provide a very limited consideration of sequencing technologies, which, in fact, are already being used for second tier testing in many states. It appears to assume that the only sequencing that might be utilized is WES or comparable comprehensive sequence analysis.

Line 186-238: I appreciate the commentary on the issues of equity and bias in interpretation, cost and potential impact on other programs. trust and transparency.

Line 282: comment on use of the CLIR worldwide dataset of NBS values that is currently being used to reduce false positive results is warranted here as an example of collaborative efforts to do just what is stated as essential by the author

 Line 303-305: This statement is unsupported and potentially false. Plus, there is no comment on state responsibility to those infants who can be identified with early onset disorders and who could benefit from treatment who would be missed if the screening was discontinued. What kinds of suits might arise from children who are seriously affected but who could have been picked up where it not for the occurrence of later onset variants of the condition?

Line 314 and following: consider acknowledging presentations and discussions at the ACHDNC on this topic, e.g. fragile X.

Line 326 and following: no acknowledgement of the work already done in this area, particularly in prenatal NBS education. 

Reviewer 2 Report

Thank you very much for letting me participate in the discussion regarding this interesting article prior to its publication. I honestly think that articles of these characteristics should be promoted.

The article presents real facts, based on correctly cited literature and makes an analysis following a known value framework. Then, it makes clear the author's opinion regarding the cases discussed (Pompe and ADL-X linked) and from the general perspective of the article.

The point you make regarding the discussion of the need for informed consent in new incorporated conditions seems fundamental to me, especially because, as the author mentions when referring to Pompe, some of the new incorporated pathologies suggest that what we are doing are population studies financed by health systems rather than newborn screening.

The other point that I think is important to highlight, and perhaps expand on in the discussion, is the under-representation in genetic databases of information from immigrant populations in the US, particularly from patients who come from multiple Latin American countries, which in some states constitute up to 50% of the population served, and that from their countries the request for genetic studies is very restricted due to the lack of health insurance coverage.

I think that articles like this raise a deep and necessary discussion regarding what we are doing, not only at the local level in the US, but fundamentally what path is being shown to countries that today do not have newborn screening, or that are in preliminary stages and plan to expand their programs.

I would love to see in the conclusion, which seems appropriate to the title of the article, a position regarding informed consent in different situations. Should it be requested for new conditions included in the research programs during the first years of incorporation? Should it always be requested when the study is genetic?

Reviewer 3 Report

Thanks for the reminds and alerts from the author. I want to take the advantage as a reviewer to harmonize newborn screening with public health ethics, try not to let them collide with each other.

1. Most of the new additions to newborn screening arose from new knowledge and technologies. Some went too fast and gave troubles, but some should be good. Therefore, in the abstract, it would be nice if the author can allow to change this sentence to “Some of these new and proposed additions to newborn screening entail serious public health ethical issues that need to be explored”

2. About the three examples of newborn screening with troubles, I hope the authors can add what the screening society has responded:

Krabbe disease: Knowing the trouble from the Krabbe disease screening program in New York, other screening program withheld screening for Krabbe disease.

ALD: The problem of X-ALD screening is in females. Some screening programs adjusted the cutoff of the screening that only a small portion of female carriers were picked up.

Pompe disease: The high false-positive rate of newborn screening for Pompe disease can be improved by adjusting the screening methods, or adding a molecular second-tier test.

3. For 2.5. (Sustaining newborn screening will require challenging adjustments) Molecular second-tier testing can effectively reduce the false positive rate of the biochemical tests.
